# Inorganic Arsenic Induces Elevated p53 Levels with Altered Functionality Impacting the Expression of Toll-like Receptor 3 and Other Target Genes in Immortalized Prostate Epithelial Cells

**DOI:** 10.3390/ijms26094253

**Published:** 2025-04-29

**Authors:** Nancy C. Pacheco-Castillo, Jesús Gómez-Montalvo, Vanesa Olivares-Illana, Félix Recillas-Targa, Erik J. Tokar, S. Eréndira Avendaño-Vázquez, Claudia Escudero-Lourdes

**Affiliations:** 1Laboratorio de Inmunotoxicología, Facultad de Ciencias Químicas, Universidad Autónoma de San Luis Potosí, San Luis Potosí 78210, Mexico; a212032@alumnos.uaslp.mx; 2Consorcio de RNA, Laboratorio de Metabolismo de RNA Largos y Medicina Molecular, Instituto Nacional de Medicina Genómica, Ciudad de México 14610, Mexico; a01374602@tec.mx; 3Laboratorio de Interacciones Biomoleculares y Cáncer, Instituto de Física, Universidad Autónoma de San Luis Potosí, San Luis Potosí 78210, Mexico; vanesaolivaresillana@gmail.com; 4Instituto de Fisiología Celular, Departamento de Genética Molecular, Universidad Nacional Autónoma de México, Ciudad de México 04510, Mexico; frecilla@ifc.unam.mx; 5Stem Cell Toxicology Group, Division of Translational Toxicology, National Institute of Environmental Health Science, Durham, NC 27709, USA; erik.tokar@nih.gov

**Keywords:** prostate cancer, *TLR3*, p53, *TP53*, *MDM2*, *CDKN1A*, gene expression, inorganic arsenic

## Abstract

Prostate cancer (PCa) is a major global health concern, particularly in advanced stages where chemotherapy resistance and androgen-independent tumor growth reduce survival rates to below 30%. Toll-like receptor 3 (TLR3), regulated by tumor suppressor p53, is a promising therapeutic target due to its role in tumor cell apoptosis. However, chronic exposure to inorganic arsenic (iAs), a known carcinogen, has been linked to PCa progression and reduced TLR3 expression and activation by polyinosinic/polycytidylic acid (Poly(I/C)), a synthetic ligand used in PCa immunotherapy. Here, we demonstrate that chronic sodium arsenite (NaAsO) exposure increases p53 transcript and protein levels in immortalized prostate epithelial cells. Despite this, key p53 target genes, including *TLR3*, *CDKN1A*, and *BAX*, were significantly downregulated, indicating a transcriptionally inactive p53. Chromatin immunoprecipitation (ChIP) confirmed diminished p53 binding to *TLR3* and *CDKN1A* promoters, while sequencing ruled out *TP53* mutations. A bioinformatic analysis revealed elevated *TP53* but reduced *TLR3* and *CDKN1A* in prostate adenocarcinoma, suggesting that iAs-induced oxidative stress disrupts p53 function. These findings reveal a novel mechanism by which iAs promotes PCa progression through impaired p53 activity, highlighting the need to explore post-translational and epigenetic factors affecting p53. Restoring p53 transcriptional activity may offer a therapeutic strategy for PCa patients exposed to NaAsO.

## 1. Introduction

Prostate cancer (PCa) is the second leading cause of cancer-related deaths in men worldwide, primarily affecting individuals over 50 years old [1]. In advanced stages, survival rates drop below 30%, driven by chemotherapy resistance and androgen-independent tumor growth [2]. Immunotherapies, including polyinosinic/polycytidylic acid (Poly(I/C)), which activates Toll-like receptor 3 (TLR3), have emerged as potential treatments [3]. TLR3 activation induces apoptosis in cancer cells, offering hope for advanced PCa [4].

Risk factors for PCa include (e.g., diet, weight, exposure to contaminants, age, race, and family history) elements [5,6]. Among these, chronic exposure to inorganic arsenic (iAs), a known carcinogen, significantly increases the risk of developing PCa [7]. Chronic exposure to iAs has emerged as a pressing global health concern. Millions of people worldwide are exposed to iAs through contaminated drinking water, food, air, or occupational settings—often without being aware of the long-term consequences. iAs is a well-documented environmental toxicant and carcinogen, classified as a Group 1 carcinogen by the IARC due to its widespread presence in natural and human-made environments [8]. It primarily exists as arsenate [As(V)] and arsenite [As(III)], the latter being significantly more toxic and mobile—up to 60 times more so than As(V) [9,10]. Notably, long-term sodium arsenite (NaAsO) exposure *in vitro* has been shown to transform RWPE-1 prostate epithelial cells into malignant phenotypes (resulting in a cell line called CAsE-PE) [11]. This transformation reduces *TLR3* transcript and protein levels and impairs Poly(I/C)-induced apoptosis in CAsE-PE cells [12]. These findings suggest that iAs disrupts pathways regulating *TLR3* expression, contributing to chemotherapy resistance.

Several regulators have been described for *TLR3* expression. One of them is IFN-alpha (IFN-α), which positively regulates *TLR3* expression by activating the JAK/STAT signaling pathway. Upon binding its receptor (IFNAR), IFN-α induces the formation of the ISGF3 complex (STAT1/STAT2/IRF9), which translocate to the nucleus and binds to interferon-stimulated response elements (ISRE) located approximately −30 bp upstream of the human *TLR3* promoter region. This regulation occurs independently of TLR3 activation by its ligands. Another cytokine, IL-27, has also been shown to regulate *TLR3* expression in specific contexts, such as human melanomas, where it enhances Poly(I/C)-induced tumor growth inhibition in a partially TRAIL-dependent manner [13].

Furthermore, the tumor suppressor protein p53 is a key regulator of *TLR3* expression under stress conditions. The p53 protein tetramerizes and binds to response elements in the *TLR3* promoter, regulating its transcription along with other target genes involved in apoptosis, cell cycle arrest, DNA repair, autophagy, and the tumor microenvironment [14]. This regulatory function is negatively modulated by the mouse double minute 2 homolog (MDM2), an E3 ubiquitin ligase that targets p53 for proteasomal degradation. Notably, p53 transcriptionally activates *MDM2*, establishing a feedback loop that fine-tunes p53 activity [15].

Since p53 is the main positive regulator of *TLR3* [12], the observation that prostate epithelial cells transformed with NaAsO express significantly lower levels of this receptor strongly suggests that the expression and/or functionality of p53 is compromised in these cells. This hypothesis is supported by previous studies. In one such study, a human keratinocyte cell line (HaCaT) was exposed to 1 µM sodium arsenite for 20 weeks, resulting in the generation of the HaCaT-T cell line. In these cells, the authors observed a significant decrease in p53 protein expression, while its negative regulator, MDM2, showed increased transcript and protein levels in the same cell model when compared to cells not exposed to iAs [16,17]. These findings suggest that the negative regulation of p53 expression and function predominates in cells exposed to this toxicant.

To test this hypothesis, in the present study, two cell lines were generated by chronically exposing the prostate epithelial cell lines RWPE-1 and HPrEC to NaAsO. This approach aimed to study the expression status of p53 and MDM2 proteins and to elucidate the mechanism by which iAs induces a decrease in TLR3 expression in these cells. However, as the research progressed, additional results prompted the exploration of alternative mechanisms by which iAs toxicity may affect p53 functionality.

## 2. Results

### 2.1. Exposure to NaAsO Leads to Increased p53 Expression in Prostate Epithelial Cells

To evaluate whether chronic NaAsO exposure affects the expression of key regulators, potentially involved in cellular functions and tumorigenic transformation, we performed Western blot (WB) and qPCR analyses to quantify protein levels and transcripts of p53, TLR3, and MDM2 in prostate epithelial cell lines. These analyses revealed significant differences in the expression patterns of these genes at both the transcript and protein levels (Figure 1). Transcript levels of *TP53*, *TLR3*, and *MDM2* were significantly higher in RWPE-1 cells compared to those in HPrEC cells. However, as expected, a notable difference in cellular p53 protein levels was observed, with RWPE-1 cells exhibiting lower p53 protein levels due to the immortalization method, which impacts p53 protein levels (Figure 1a).

In contrast, exposure to NaAsO for 29 weeks led to significant decreases in *TLR3* and *MDM2* transcript levels in CAsE-PE and HAsE-PE cells compared to those in their parental cell lines (Figure 1b,c). Surprisingly, both transcript and protein levels of p53 were significantly increased in both NaAsO-exposed cell lines (CAsE-PE and HAsE-PE) compared to the non-exposed parental cells (RWPE-1 and HPrEC, respectively) (Figure 1a). This increase in p53 expression was observed as early as the initial weeks of NaAsO exposure and remained consistent until the maximum proposed exposure time was reached (Appendix A. In conclusion, chronic exposure to NaAsO altered the expression of p53, TLR3, and MDM2 in prostate epithelial cells. RWPE-1 cells exhibited increased transcription levels of these genes compared to HPrEC cells, whereas prolonged exposure significantly decreased *TLR3* and *MDM2* expression in CAsE-PE and HAsE-PE cells. In contrast, p53 exhibited a sustained increase at both the RNA and protein levels in the exposed cells, suggesting that chronic NaAsO exposure significantly impacts the regulation of key genes in prostate epithelial cells.

### 2.2. Consistent Findings with External Datasets

To further explore the unexpected increase in p53 expression in NaAsO-exposed cells, additional bioinformatic analyses were conducted. Using RNA-seq data from [18], which analyzed gene expression in CAsE-PE and RWPE-1 cells, we compared the differential expression of genes of interest. Expression levels were calculated in terms of FPKM (Fragments Per Kilobase of Exon per Million Reads), a metric used to normalize and compare transcript abundance. Consistent with our findings, the data in [18] showed a significant increase in p53 expression in CAsE-PE cells transformed by NaAsO, compared to that in naïve RWPE-1 cells. Additionally, as in our study, the expression of p53 target genes such as *TLR3*, *MDM2*, *CDKN1A*, *BAX*, and *IGFBP3* [19] showed a decreasing trend in CAsE-PE cells (Figure 2). Interestingly, two other p53-regulated genes, *GADD45A* and *ZNF385A*, showed increased expression (Appendix A). In conclusion, the bioinformatics analysis confirmed an increase in p53 expression in CAsE-PE cells exposed to NaAsO, consistent with the findings of [18]. Additionally, a decrease in the expression of its target genes was observed, except for *GADD45A* and *ZNF385A*, which exhibited an increase, suggesting a possible alteration in p53 transcriptional regulation or a differential response of its target genes.

### 2.3. Tissue-Specific Insights from Bioinformatic Analysis

To determine whether the observed effects were specific to NaAsO exposure or a general feature of prostate cancer (CaP), OncoDB, a bioinformatic tool based on The Cancer Genome Atlas (TCGA), was used. This tool provides graphical representations of gene expression in cancer across 33 tissue types. Prostate adenocarcinoma was selected to analyze the expression levels of *TP53*, *TLR3*, *MDM2*, and *CDKN1A*. Consistent with our experimental data, OncoDB revealed a slight but significant increase in the expression of *TP53*, and a decrease in the expression of *TLR3* and *CDKN1A* in prostate cancer tissues (Figure 3). These results suggest that the alterations in these pathways are not exclusive to NaAsO exposure but may be a broader feature of prostate cancer biology.

### 2.4. Comparative Gene Expression Analysis in Non-Prostatic Tumors Suggests Context-Specific p53 Dysfunction

To gain a broader context on our findings, we explored public RNA-seq data from TCGA to evaluate whether the uncoupling between *TP53* expression and its downstream targets (e.g., *TLR3*, *CDKN1A*, and *MDM2*) observed in arsenic-transformed prostate epithelial cells also occurs in other tumor types. We selected adrenocortical carcinoma (ACC) and chromophobe kidney carcinoma (KICH) as comparative models based on their anatomical and developmental proximity to the prostate. All three organs—the adrenal gland, kidney, and prostate—derive from intermediate mesodermal lineages and share common regulatory pathways during urogenital development, making them suitable for exploring whether the observed dysfunction in p53 is tissue-specific.

Using RNA-seq data from OncoDB, we analyzed the expression of *TP53* and its target genes in tumor versus normal tissue samples. In both ACC and KICH, *TP53* expression levels were positively correlated with those of *TLR3* and *CDKN1A*. For example, in ACC, the expression of *TP53*, *TLR3*, and *CDKN1A* tended to increase in tumor samples compared to their non-tumor counterparts. Similarly, in KICH, decreases in *TP53* expression were accompanied by reductions in *TLR3* and *CDKN1A* levels (Figure 4a,b). These patterns suggest that in these non-prostate tumor types, p53 retains its ability to activate downstream transcriptional targets, even when its expression is elevated.

This behavior contrasts sharply with what we observed in prostate epithelial cells chronically exposed to NaAsO, where increased p53 expression coincided with a loss in transcriptional activity, as evidenced by the reduced expression of canonical targets such as *TLR3* and *CDKN1A*. These comparative observations reinforce the idea that p53 dysfunction in response to iAs exposure may involve prostate-specific regulatory mechanisms, possibly influenced by the tissue microenvironment or lineage-specific cofactors that alter p53’s ability to activate target genes. While preliminary, these data underscore the need to consider tissue context when evaluating functional p53 activity in cancer.

### 2.5. NaAsO Reduces the Binding Ability of p53 to the Promoter Regions of Target Genes

The transcriptional capacity of p53 relies on two key processes, its tetramerization and binding of the tetrameric conformation to DNA at response elements in the promoters of its target genes. In this study, the ability of p53 to bind to the promoter regions of the *TLR3* and *CDKN1A*, which encodes the p21 protein, was analyzed. Previous studies, including [18], reported reduced *TLR3* and *CDKN1A* expression in CAsE-PE cells exposed to NaAsO, further suggesting an impairment of p53 functionality. To assess p53 binding, a chromatin immunoprecipitation (ChIP) assay was performed, with chromatin extracted from CAsE-PE and HAsE-PE cells. Figure 5a,b illustrate the location and sequences of the oligonucleotides used for amplifying the DNA regions immunoprecipitated with anti-p53 IgG in the ChIP assay. The results, depicted in Figure 5c,d, revealed a significant reduction in p53 binding to *TLR3* and *CDKN1A* promoter regions in cells chronically exposed to NaAsO, compared to their non-exposed controls. These findings highlight a direct effect of NaAsO on p53 functionality, potentially contributing to the dysregulation of critical genes in cellular stress responses.

### 2.6. Reduced p53 Binding Cannot Be Attributed to Mutations in the TP53 Gene

Mutations in p53 are frequently observed in various types of cancer and are known to impair its transcriptional activity and tumor-suppressor function. To investigate whether chronic iAs exposure results in mutation, deletion, or insertion in *TP53* coding regions that could explain the observed reduction in p53 binding, we performed Sanger sequencing of cDNA from CAsE-PE and HAsE-PE cells. Figure 6a outlines the *TP53* gene structure and presents the oligonucleotide pairs used to amplify specific regions of the *TP53* coding sequence: primer pair 1–360 targets the transactivation domains I and II (TAI, TAII) and proline-rich region (PP) domains; primer pair 313–833 spans the DNA binding domain (DBD); and primer pair 879–1194 covers the oligomerization domain (OD) and trans-activation (TD) domains. Sequencing results revealed no mutations in the analyzed regions of *TP53*, including the DBD and TD or C-terminal domains. Figure 6b shows the corresponding p53 protein functional domains, TA (I and II), PP, DBD, OD, and TD. These findings suggest that the observed reduction in p53 binding to DNA in iAs-exposed cells is not caused by mutations in the *TP53* coding region but may instead result from other factors, such as post-translational modifications or altered cofactor interactions.

## 3. Discussion

Human exposure to inorganic arsenic (iAs) from various sources has been identified as a risk factor for PCa development [20]. Immunotherapy based on TLR3 activation using synthetic dsRNA ligands like Poly(I/C) has shown promise in advanced PCa stages [21]. Importantly, our research demonstrated that chronic sodium arsenite (NaAsO) exposure to prostate epithelial cells in vitro reduces *TLR3* expression, impairing its activation by Poly(I/C). This study aimed to explore whether this downregulation of TLR3 is due to a reduction in the expression or functionality of p53 protein, a key positive transcriptional regulator of most TLRs [22]. Supporting evidence includes reports showing that keratinocytes chronically exposed to NaAsO exhibit reduced *TP53* transcript and protein levels while increasing *MDM2* expression [23].

In our study, two prostate epithelial cell lines, RWPE-1 and HPrEC, exposed to NaAsO displayed reduced TLR3 expression. Surprisingly, however, NaAsO exposure significantly increased p53 transcript and protein levels compared to those in untreated cells. Interestingly, RWPE-1 cells, which exhibit lower basal p53 protein levels due to their immortalization method, showed earlier p53 overexpression than HPrEC cells, highlighting a differential susceptibility to iAs between the two cell lines.

Our findings align with those of [18]. This study also reported increased p53 transcript and protein levels but decreased TLR3 expression in CAsE-PE cells. Interestingly, other genes regulated by p53, including *MDM2*, *CDKN1A*, *BAX*, and *IGFBP3*, also showed decreased expression, suggesting that iAs exposure induces transcriptionally dysfunctional p53. However, genes such as *GADD45A* and *ZNF385A*, which are also p53 targets, showed increased expression [18].

This discrepancy may be due to alternative regulatory mechanisms. While p53 is the primary regulator of *CDKN1A*, *BAX*, *MDM2*, *GADD45A*, and *ZNF385A* [24,25], additional factors can influence the expression of *GADD45A* and *ZNF385A*. For instance, *GADD45A*, a stress-induced gene crucial for DNA repair, cell cycle arrest, and apoptosis, is also regulated by *BRCA1* [26,27]. Similarly, *ZNF385A*, a transcription factor that modulates p53 target genes, responds to cellular stress and senescence signals and can differentially activate *CDKN1*.

In the absence of functional p53, proteins such as p63 and p73 may compensate for its loss [28]. Additionally, other transcription factors, including members of the FOXO family, NF-κβ and the AP-1, may regulate genes involved in DNA repair and apoptosis, such as *GADD45A* [29]. Similarly, HIF-1α and effectors of the Wnt/β-catenin pathway have been shown to influence the expression of *ZNF385A*.

On the other hand, iAs exposure has the potential to induce p53 phosphorylation at specific residues, which can modulate its interactions with DNA and transcriptional cofactors. p53 is phosphorylated at serine or threonine residues, particularly in response to oxidative damage [30]. These modifications can selectively enhance or suppress the transcriptional activation of specific target genes, depending on the phosphorylated site and the cellular context. For instance, phosphorylation at Ser15 and Ser20 stabilizes p53, promoting its accumulation and activation of apoptosis-related genes, whereas modifications at other sites may interfere with DNA binding or the recruitment of transcriptional cofactors [31]. This differential phosphorylation of p53 could explain the selective activation of genes like *GADD45A* and *ZNF385A*, while genes such as *CDKN1A* and *MDM2* remain unresponsive. Therefore, post-translational modifications of p53, particularly phosphorylation, appear to play a crucial role in determining its functional activity in iAs-exposed prostate epithelial cells.

The observed p53 overexpression in prostate epithelial cells may result from oxidative stress and DNA damage induced by iAs exposure. Arsenite radicals bind to sulfhydryl groups in mitochondrial proteins, disrupting the electron transport chain, reducing ATP production and increasing superoxide and hydrogen peroxide levels [32]. Additionally, iAs activates *NADPH* oxidase and depletes glutathione, further compromising antioxidant defenses [33].

While oxidative stress is known to elevate p53 expression in various tissues, our transcriptomic analysis—using two different independent datasets—revealed a consistent increase in p53 levels specifically in prostate epithelial cells and prostate associated tissues. This observation suggests that a reduced antioxidant capacity, greater DNA damage, inflammation, and cellular injury make prostate-associated cells particularly susceptible to oxidative stress, unlike other non-prostatic cancers [34,35].

In the prostate epithelial cell lines exposed to NaAsO in this study, the sustained overexpression of p53 may have resulted from oxidative and nitrosative stress induced by iAs. This stress damages key cellular components, including lipids, proteins, and DNA [36,37]. Additionally, iAs depletes antioxidants such as glutathione and vitamins C and E, while inhibiting enzymes like catalase, superoxide dismutase (SOD), and glutathione peroxidase, which are essential to neutralize reactive oxygen species (ROS) and reactive nitrogen species (RNS), leaving cells more susceptible to oxidative damage [38].

The increased p53 expression observed in human prostate cancer tissues from the OncoDB database—unrelated to iAs exposure—could be explained by the detrimental effects of testosterone on the antioxidant capacity of prostate tissue. This effect appears to be specific to prostate tumors and does not occur in non-prostate cancers. Testosterone is known to stimulate free radical production [39]. With aging, the antioxidant defenses in prostate tissue naturally decline, making it more vulnerable to damage by ROS and RNS [17]. These reactive species not only exacerbate oxidative stress but also stimulate the expression of pro-inflammatory cytokines, which further amplify the generation of reactive molecules and oxidative damage [40].

In this work, we established that iAs exposure significantly elevates p53 expression in RWPE-1 and HPrEC prostate epithelial cells. However, the notable decrease in the expression of key p53 transcriptional targets, such as the genes encoding TLR3, p21, and BAX, strongly suggests that the transcriptional activity of p53 is impaired in these cells. To test this hypothesis, we performed ChIP assays to assess p53’s ability to bind to the promoters of the *TLR3* and *CDKN1A* genes in cells chronically exposed to NaAsO compared to non-exposed cells. The results confirmed a significant reduction in p53 binding capacity to the regulatory regions of these genes, indicating that NaAsO exposure compromises the functional activity of p53.

To explore potential mechanisms underlying this reduced binding capacity, we sequenced the *TP53* coding region in CAsE-PE and HAsE-PE cells. The analysis revealed no mutations in the DNA-binding domain (DBD) or tetramerization domain, ruling out coding regions mutations as the cause of impaired p53 function. This suggests that the reduced p53 binding observed in iAs-exposed cells may result from post-translational modifications, such as phosphorylation, acetylation, or sumoylation, which can alter p53’s DNA-binding affinity or its interaction with essential transcriptional cofactors.

Other potential mechanisms not explored in this study include epigenetic modifications, such as DNA methylation or histone modifications, which could restrict p53 access to its target gene promoters. Additionally, disruptions in pathways like ATM/ATR, that regulate p53 activation, may also play a role [41]. Understanding these mechanisms would be crucial for uncovering how iAs disrupts p53 function and its downstream regulatory network. Future studies should further investigate iAs-induced effects on p53 tetramerization, thiol interactions, and its recruitment of transcriptional machinery, as these could provide valuable insights into the mechanisms driving the impaired p53 activity observed in iAs-exposed prostate epithelial cells.

Our findings suggest that chronic inorganic arsenic (iAs) exposure induces a paradoxical increase in p53 transcript and protein levels while impairing its transcriptional activity. A plausible mechanism involves iAs-mediated oxidative stress, driven by arsenite (As^3+^), which disrupts mitochondrial function and promotes the accumulation of reactive oxygen species (ROS) such as superoxide and hydrogen peroxide [42,43]. These ROS not only cause DNA damage—thereby stabilizing p53—but may also impair the enzymatic machinery responsible for its post-translational modifications. Under physiological conditions, modifications such as phosphorylation or acetylation enhance p53 stability and transcriptional competence. However, chronic oxidative stress may oxidize critical cysteine residues in p53 and its modifying enzymes (e.g., ATM/ATR, p300/CBP), thereby blocking functional activation [44,45]. This model could explain why, in arsenic-transformed prostate cells, p53 overexpression fails to translate into the activation of canonical target genes such as *TLR3* and *CDKN1A*. Importantly, this dysfunction occurs in the absence of *TP53* mutations, suggesting a mechanism of functional inactivation that challenges the conventional paradigm equating p53 abundance with activity. Further studies are needed to directly evaluate post-translational modifications of p53 under chronic iAs exposure, which could validate this proposed mechanism and identify targets for the therapeutic reactivation of p53 function in prostate cancer.

Although our study provides novel insights into the functional disruption of p53 following chronic iAs exposure in prostate epithelial cells, it is important to acknowledge certain limitations. Our findings are based on non-tumorigenic immortalized prostate epithelial cell lines (RWPE-1 and HPrEC), which offer a controlled in vitro environment but may not fully capture the complexity of in vivo systems. In the tumor microenvironment, additional factors such as stromal cell interactions, immune cell infiltration, cytokine gradients, and systemic responses (e.g., hormonal or metabolic signals) may influence both p53 activity and TLR3 expression, potentially modulating the outcomes observed here. Moreover, widely used prostate cancer cell lines such as LNCaP, LAPC4, DU145, and PC3 each present different combinations of androgen receptor (AR) status and p53 functionality—ranging from wild-type to mutated or null—highlighting the heterogeneity of prostate tumors [46,47]. While these models are invaluable for studying various aspects of disease progression, our use of non-transformed epithelial cells provides a distinct advantage in dissecting early molecular events triggered by environmental carcinogens like iAs. Future studies incorporating patient-derived organoids or in vivo models will be necessary to validate the broader relevance of our findings and to explore how microenvironmental factors shape the p53–TLR3 axis under chronic toxicant exposure.

Our study uncovered a novel mechanism of functional p53 disruption triggered by environmental toxicants. This phenomenon may underlie aspects of therapy resistance and immune evasion in prostate cancer, where p53 is present but inactive. The next critical milestone is to identify the specific post-translational modifications that drive this altered p53 behavior. Future work will focus on dissecting phosphorylation, acetylation, and oxidation states of p53 in iAs-transformed cells, along with the roles of upstream regulators such as ATM, ATR, and p300. These efforts may uncover actionable targets for therapeutic reactivation of wild-type p53 in prostate tumors rendered functionally silent by environmental stressors.

## 4. Materials and Methods

### 4.1. Cell Models and Cell Culture

Two non-tumorigenic prostate epithelial cell lines were used: RWPE-1, immortalized via transfection with a gene from human papillomavirus 18 (HPV-18) [48], and HPrEC cells, which were immortalized through c-Myc overexpression [49]. These cells exhibit different levels of p53 expression and functionality. Specifically, the RWPE-1 cell line was immortalized by inserting the gene encoding the HPV-18 E7 protein, which targets the protein retinoblastoma (pRb) and reduces E2F. Although E7 does not directly target p53, its effect on pRb and E2F can modify p53 levels. However, in RWPE-1 cells, p53 remains present and functional, without affecting typical adherent characteristics, growth, or cell morphology. In contrast, HPrEC cells were immortalized by c-Myc oncogene overexpression which does not affect p53 levels, maintaining typical epithelial morphology and growth.

RWPE-1 cells were exposed to 5 µM NaAsO for 29 weeks, resulting in the cancerous CAsE-PE cell line. CAsE-PE cells exhibit an increased expression of markers associated with survival, invasiveness, drug resistance, and tumor aggressiveness, as confirmed by xenograft studies [50]. Similarly, HPrEC cells were exposed to 3 µM NaAsO (Sigma-Aldrich, St. Louis, MO, USA, CAS No. 7784-46-5) continuously for 29 weeks, with medium replacement every 48 h. This concentration was selected based on preliminary viability assays (Appendix A), which showed that viability remains above 50%, allowing for sustained stress without complete cell loss. This dosage was consistent with previous transformation protocols, such as that of [51], in which chronic iAs exposure led to the malignant transformation of RWPE-1 cells. Our objective was to model chronic exposure, which better reflects long-term environmental arsenic exposure in human populations. Prior studies have indicated that prolonged iAs exposure over several months is necessary to recapitulate the molecular alterations observed in arsenic-associated malignancies. Accordingly, we selected week 29 as a representative late time point to capture the cumulative effects of sustained iAs exposure on prostate epithelial cells, resulting in the stable HAsE-PE cell line.

For routine maintenance, RWPE-1 and CAsE-PE cell lines were cultured in serum-free keratinocyte medium (KSFM; GIBCO/ThermoFisher Scientific, Waltham, MA, USA) supplemented with 50 µg/mL bovine pituitary extract (BPE) and 50 µg/mL epidermal growth factor (EGF) without antibiotics. HPrEC and HAsE-PE cells were cultured in RPMI medium (Roswell Park Memorial Institute) supplemented with 200 mM L-glutamine and 10% fetal bovine serum (FBS), without antibiotics. All cell lines were incubated at 37 °C in a humidified atmosphere with 5% carbon dioxide and were subcultured upon reaching 80–90% confluence.

### 4.2. NaAsO Toxicity Assays for HPrEC Cells

To evaluate the toxicity of NaAsO on HPrEC cells, 1 × 10^4^ cells were seeded per well in 96-well plates containing RPMI medium supplemented with 10% FBS. The cells were exposed in triplicate to NaAsO at concentrations of 1, 2, 3, 4, and 5 µM. After 24 h of exposure, NaAsO toxicity was assessed using a commercial CellTiter 96^®^ AQueous Non-Radioactive Cell Proliferation Assay (Promega, Madison, WI, USA) following the manufacturer’s instructions. Absorbance was measured at 490 nm using an ELISA plate reader.

### 4.3. Analysis of TLR3, p53, and MDM2 Protein Expression by Western Blot (WB)

To determine the effect of chronic NaAsO exposure on TLR3, p53, and MDM2 expression, RWPE-1, CAsE-PE, HPrEC, and HAsE-PE cells were cultured to 85% confluence in triplicate, washed once with PBS, and collected by trypsinization. Total protein was extracted on ice using a RIPA buffer (50 mM Tris, pH 7.5, 150 mM NaCl, 1 mM EDTA pH 8.0, 0.1% NP40, 0.1% sodium deoxycholate, 0.1% SDS) supplemented with a Complete 1X protease inhibitors cocktail (Roche, Indianapolis, IN, USA), PMSF, and NaF. The protein concentration was determined using a Bradford assay, and samples were standardized to 4 µg/µL for each sample to load a total amount of 80 µg/well in a volume of 20 µL. Proteins were resolved by SDS-PAGE on polyacrylamide gels (8%, 10%, or 15%) at 60 V for 25 min, followed by 80 V for approximately 90 min. After electrophoresis, proteins were then transferred to polyvinylidene difluoride (PVDF) membranes at 300 mA for 2 h in TBST buffer. Membranes were blocked overnight at 4 °C with 5% Blotto non-fat milk (Chem Cruz, Santa Cruz Biotechnology, Dallas, TX, USA), #cat sc-2324) in 1X TPBS and incubated with the primary antibodies anti-TLR3 (1:200, Abcam, Waltham, MA, USA, CB20 0AX), anti-MDM2 (1:500), anti-p53-DO1 (1:1000), or anti-β-actin-HRP (1:5000, Santa Cruz Biotechnology, Dallas, TX, USA, sc-126). Secondary antibodies (α-mouse or α-rabbit HRP-conjugated, Santa Cruz Biotechnology, Dallas, TX, USA, sc-2357 and sc-358914) were incubated for 2 h at room temperature. Membranes were washed three times with 1X TBS and visualized using an HRP chemiluminescent substrate (Luminol^®^, GE Healthcare Technologies, Chicago, IL, USA) in a Biorad photodocumentation system (light source 7729) (Hercules, CA, USA). The expression β-actin was used as a loading control.

### 4.4. Expression Analysis by q-PCR

The four cell lines were cultured to ~80% confluence and collected by trypsinization (0.025% trypsin). RNA was extracted from the cell pellets using TRIzol^®^ (Invitrogen, Thermo Fisher Scientific, Carlsbad, CA, USA, 15596026), and its purity and concentration were assessed using a Nanodrop spectrophotometer at λ260/280 nm (acceptable range: 1.8–2.0). RNA samples were adjusted to 2.5 µg/mL and stored at −80 °C.

cDNA synthesis was performed using 2.5 µg of RNA in a reaction containing an Oligo (dT) anchor primer (100 ng), ultrapure water (7.5 µL), 1X First Standard Buffer (DTT 0.1 M), and M-MLVRT (1 µL) for a final volume of 20 µL. The reverse transcription protocol included 65 °C for 5 min, 37 °C for 60 min, and 75 °C for 5 min.

Quantitative PCR (q-PCR) was performed using 100 ng of cDNA per reaction to analyze the expression of *TLR3*, *TP53*, and *MDM2*. Each reaction included SYBR Green solution, (Applied Biosystems, Foster City, CA, USA) primer (Appendix A), and ultrapure water in a final volume of 10 µL. *GAPDH* was used as the reference gene, and the thermal cycling protocol was 95 °C for 10 min, followed by 40 cycles of 95 °C for 2 min, 40 °C for 15 s, and 72 °C for 1 min. The relative expression was calculated using the 2^−ΔΔCt^ method.

### 4.5. Chromatin Immunoprecipitation Assay (ChIP/q-PCR)

To evaluate the binding capacity of p53 to the promoters of two of its target genes, *TLR3* and *CDKN1A*, a ChIP-qPCR assay was performed. Chromatin fixation was initiated by incubating cultures of each cell line with 11% formaldehyde, followed by quenching the reaction with 2.5 M glycine. The cells were washed with cold 1X PBS and resuspended in cell lysis buffer (10 mM Tris-HCl, pH 7.5, 10 mM NaCl, 0.3% NP-40, Complete 1X protease inhibitor cocktail, and distilled H_2_O). The suspension was incubated on ice and homogenized by inversion to release nuclei.

The nuclei were then resuspended in a nuclear lysis buffer (50 mM Tris-HCl, pH 7.5, 10 mM EDTA, 1% SDS, Complete 1X protease inhibitor cocktail, and distilled H_2_O) and incubated on ice. The chromatin was sheared by sonication using an ultrasonic disruptor (TOMY, Tokyo, Japan) with 10 pulses of 30 s at 35% amplitude, with 1 min intervals between pulses, for a total of 5 min. The lysate was centrifuged at 4 °C for 10 min at 2400 rpm to remove debris.

The resulting chromatin solution was precleared by incubation with Staphylococcus aureus protein A (Pansorbin, #507862; Calbiochem, San Diego, CA, USA). The precleared chromatin was incubated overnight with rotation at 4 °C in the presence of either an anti-p53 mouse monoclonal antibody or mouse IgG (as a negative control). After immunoprecipitation, the samples were centrifuged for 10 min at 2400 rpm at 4 °C. The chromatin was washed multiple times with cold 1X PBS, and the crosslinking was reversed by treatment with proteinase K. DNA was extracted from the immunoprecipitated chromatin using the phenol-chloroform method. The purified DNA was then analyzed by real-time PCR using I-taq polymerase (enzyme mix + SYBR Green) and forward/reverse oligonucleotides specific for *hTLR3* and *hCDKN1A*. Each reaction was performed with 50 ng/µL of DNA (from RWPE-1, CAsE-PE, HPrEC, or HAsE-PE cells) in a final volume with distilled H_2_O. The thermal cycling conditions were 95 °C for 3 min, followed by 95 °C for 10 s, 57 °C for *TLR3* or 60 °C for *CDKN1A* for 40 s, and a final melt curve analysis in the thermal cycler. The sequences of the specific oligonucleotides used are presented in Appendix A.

### 4.6. Sequencing

Sequencing was performed to identify possible mutations in the coding sequence of the *TP53* gene in cell lines chronically exposed to NaAsO, specifically CAsE-PE and HAsE-PE. For this purpose, cDNA obtained from reverse transcription was used along with three different primer pairs that collectively covered the entire transcript of the *TP53* gene (Appendix A). PCR amplification was carried out using the Pfu DNA polymerase (Thermo Scientific™ DNA polymerase) due to its high fidelity and thermal stability, ensuring the accurate replication of the coding sequence.

The PCR amplification products from each primer set were sequenced using the Sanger sequencing method. The resulting sequences were analyzed to identify mutations by comparing them against the reference *TP53* gene sequence from Ensembl (ENST00000269305, *TP53*-201; location: Chr 17: 7,668,421-7,687,490). The analysis was performed using the FinchTV software application (version 1.4.0; https://digitalworldbiology.com/FinchTV accessed on 22 October 2022), which allows visualization, editing, and alignment of chromatograms. Additionally, a BLAST analysis was conducted to compare the nucleotide sequences against the reference sequence.

### 4.7. Transcriptomic Data Analysis

Under the accession number PRJNA514268, we consulted the differential gene expression analysis data originally published in Appendix A titled “RNA-seq DEGs by CuffDiff for transformed CAsE-PE cells compared to control RWPE-1 cells”. This table contains the statistical values derived from the Cufflinks analysis of differentially expressed genes (DEGs) between CAsE-PE and RWPE-1 cells [18]. A gene was considered differentially expressed if its q-value was less than 0.05. Cufflinks was used to calculate expression levels in terms of FPKM (Fragments Per Kilobase of Exon Per Million Reads), allowing for normalization and comparison of transcript variants. For genes with multiple transcript variants, the statistical values corresponding to the most highly expressed variant were selected as representative for that gene.

### 4.8. OncoDB Database Analysis

To evaluate gene expression changes in prostate tumors and compare them to other cancer types, we used the publicly available database OncoDB (https://oncodb.org) [34], accessed in February 2023. The Expression Analysis module was used to retrieve normalized RNA-seq data (TPM) from The Cancer Genome Atlas (TCGA) for Prostate Adenocarcinoma (PRAD) and for two additional tumor types: Adrenocortical Carcinoma (ACC) and Kidney Chromophobe (KICH). For each gene of interest (*TP53*, *TLR3*, *CDKN1A*, and *MDM2*), expression levels in tumors versus normal tissues were compared using the built-in statistical tools provided by OncoDB, which implemented Student’s *t*-test for a differential expression analysis. Genes were considered differentially expressed when *p*-values were <0.05. Fold change (FC) values were calculated as log_2_ (tumor/normal) and reported to support the interpretation of expression trends. Box plots were generated within the OncoDB platform, and the underlying numerical data were exported for figure construction in GraphPad Prism 5 to produce high-resolution visuals. These plots display expression distributions across tumor and normal tissues, enabling direct visual comparison across tumor types. This database-driven approach allowed us to systematically examine whether the transcriptional relationships observed in prostate epithelial cells under iAs exposure are also present—or differ—in other malignancies.

### 4.9. Statistical Analysis

The statistical analysis for all results obtained from the Western blot (WB), q-PCR, and ChIP assays was performed using GraphPad Prism. A two-way ANOVA was used for group comparisons and a Bonferroni post hoc test was applied to assess differences between treated and untreated cells. Significance thresholds were defined as follows: ns: not significant, * *p* ≤ 0.05, ** *p* ≤ 0.005, and *** *p* ≤ 0.0005.

For the transcriptomic analysis from the dataset by Merrick B. et al. (2019) [18], RNA-seq data were processed using the Cufflinks tool to calculate expression levels in terms of FPKM (Fragments Per Kilobase of Exon Per Million Reads). This normalization enables the comparison of different gene expression levels. Genes were considered differentially expressed if they exhibited a q < 0.05.

For the OncoDB results, RNA-seq data were normalized by TPM (Transcripts Per Million) and visualized using box plots. Genes were classified as significantly differentially expressed if they showed a *p*-value < 0.05.

## Figures and Tables

**Figure 1 ijms-26-04253-f001:**
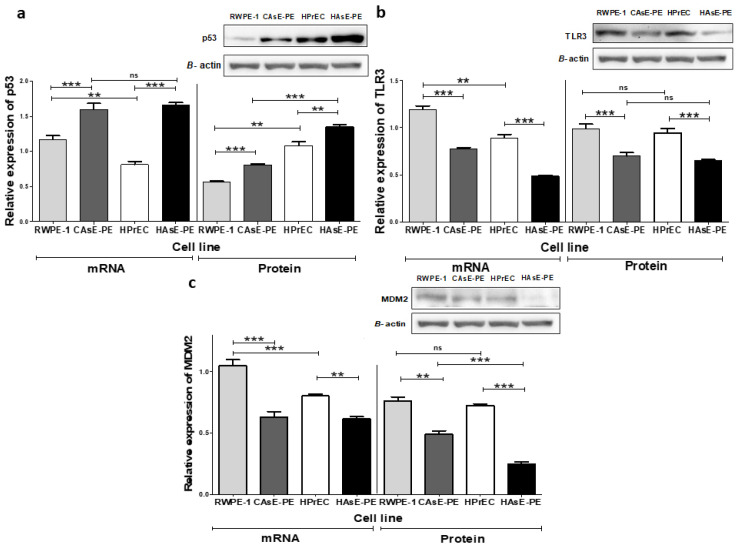
Effect of NaAsO on mRNA and protein levels. (**a**) Relative expression of *TP53* mRNA and protein; (**b**) relative expression of *TLR3* mRNA and protein; (**c**) relative expression of *MDM2* mRNA and protein levels. Data represent *n* = 3 (mean ± SD) of biological replicates. The analysis was performed using a two-way ANOVA followed by the Bonferroni post hoc test. Significance indicators: *ns*: not significant, ** *p* ≤ 0.005, *** *p* ≤ 0.0005.

**Figure 2 ijms-26-04253-f002:**
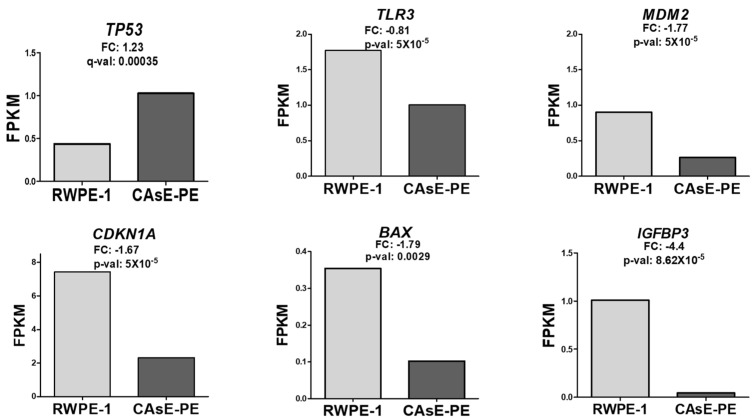
Effect of NaAsO on the expression of p53 target genes (*TLR3*, *MDM2*, *CDKN1A*, *BAX*, and *IGFBP3*) in RWPE-1 and CAsE-PE cells. Data are from [18]. Expression was calculated in FPKM. FC: log2 (fold change); adjusted *p*-value (q-val).

**Figure 3 ijms-26-04253-f003:**
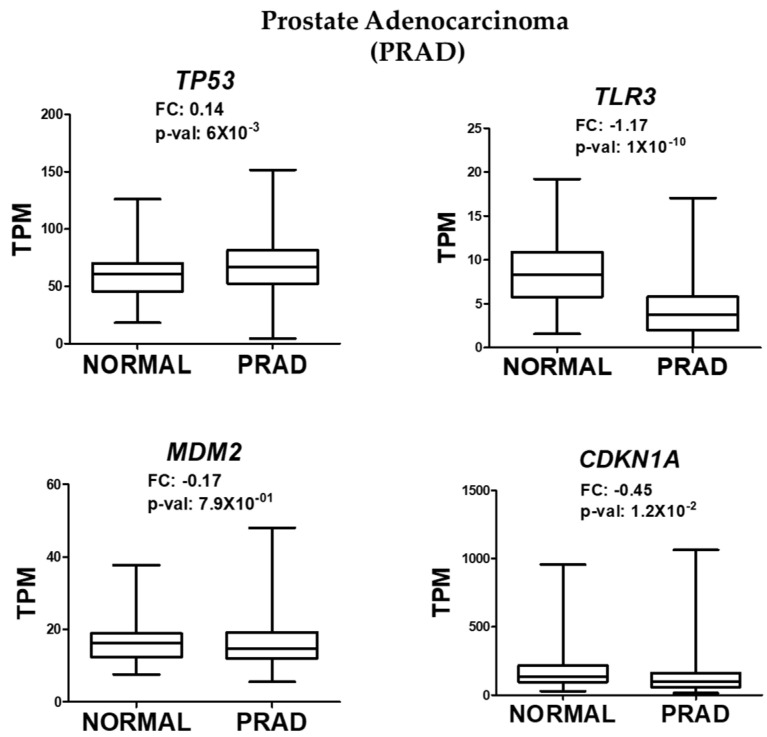
RNA-seq analysis of *TP53*, *TLR3*, *MDM2*, and *CDKN1A* expression in prostate adenocarcinoma. Data were obtained from OncoDB and normalized by TPM (transcripts per million), with significant values defined as *p*-value < 0.05.

**Figure 4 ijms-26-04253-f004:**
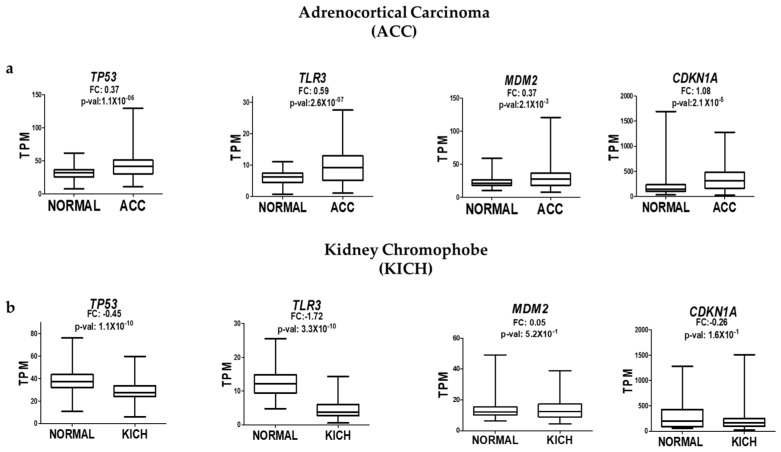
Analysis of *TP53*, *TLR3*, *MDM2*, and *CDKN1A* expression in adrenocortical carcinoma (ACC) (**a**) and chromophobe kidney carcinoma (KICH) (**b**) based on RNA-seq data from OncoDB. Expression levels were normalized as TPM (transcripts per million), with significant values defined as *p*-value < 0.05.

**Figure 5 ijms-26-04253-f005:**
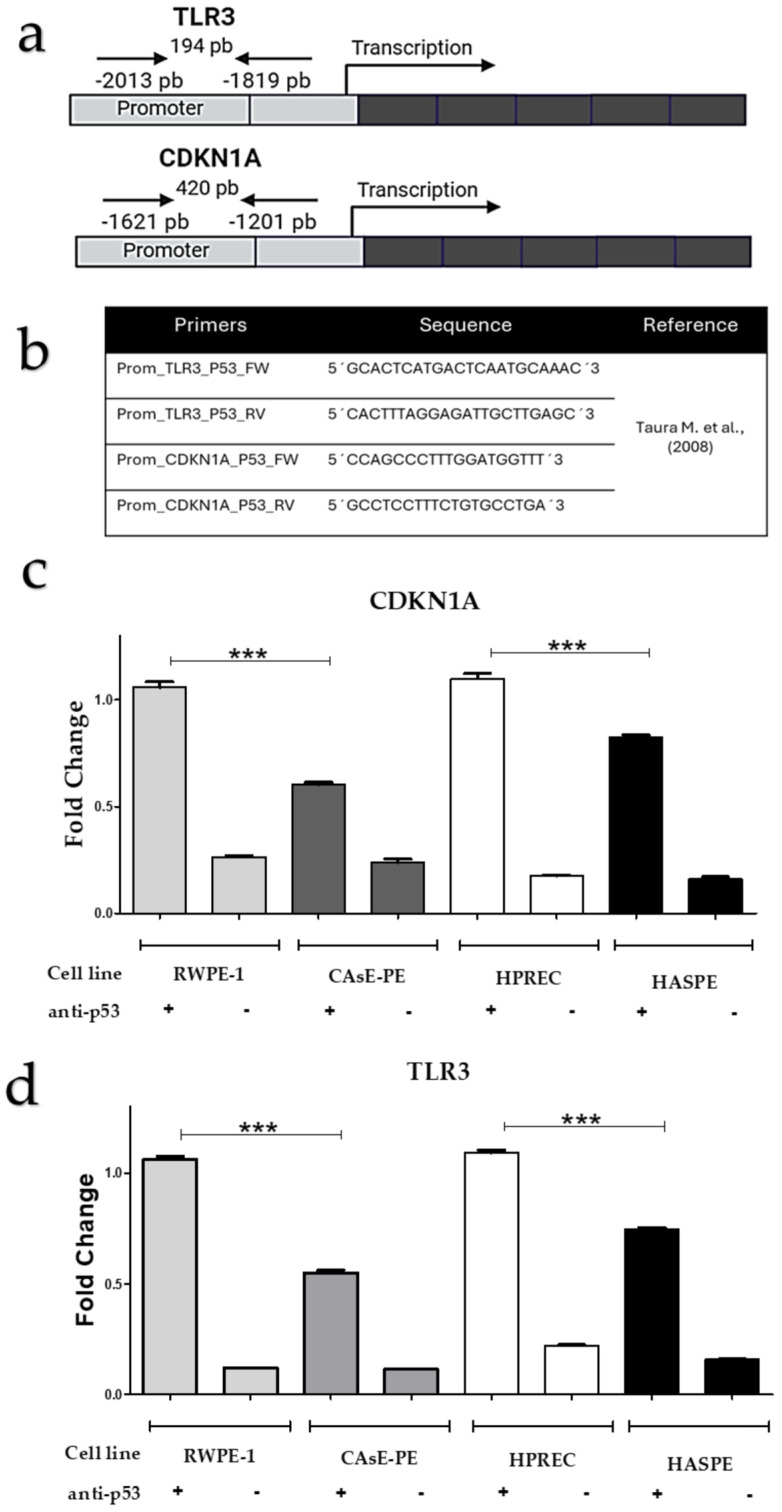
Binding of p53 to the promoter regions of the *TLR3* and *CDKN1A* genes. (**a**) Diagram showing the location of oligonucleotides in the promoter regions of *TLR3* and *CDKN1A*, respectively. (**b**) Sequences of oligonucleotides used for amplification of the promoter regions in the ChIP assay (adapted from [14]). (**c**,**d**) ChIP results for p53 binding to the promoter regions of *TLR3* and *CDKN1A* in CAsE-PE and HAsE-PE cells compared to their respective controls not exposed to NaAsO (RWPE-1 and HPrEC). Data are presented as means ± standard deviations from triplicate experiments. Statistical analysis was performed using two-way ANOVA followed by Bonferroni post hoc test. Significance levels: *** *p* ≤ 0.0005.

**Figure 6 ijms-26-04253-f006:**
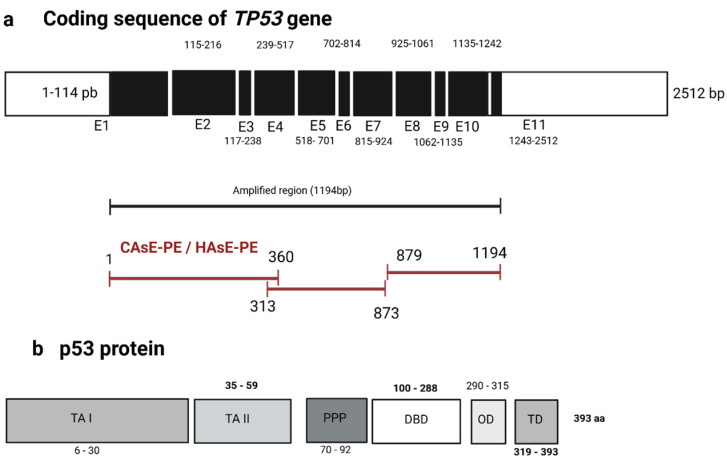
Sequencing of cDNA from the *TP53* gene in CAsE-PE and HAsE-PE cell lines exposed to NaAsO. (**a**) Schematic representation of the cDNA coding sequence of the *TP53* gene, regions and sizes of the fragments amplified from the cDNA of CAsE-PE and HAsE-PE cell lines. (**b**) Diagram of the p53 protein, highlighting its functional domains: trans-activation domains (TA I, and II), proline-rich region (PPP), DNA binding domain (DBD), oligomerization domain (OD), and amino-terminal trans-activation domain (TD).

## Data Availability

This study did not generate new datasets. The analysis was conducted using publicly available RNA-seq differential gene expression data under the accession number PRJNA514268. Specifically, we reanalyzed data from Appendix A, titled “RNA-seq DEGs by CuffDiff for transformed CAsE-PE cells compared to control RWPE-1 cells”.

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
