# Peer review of "Inorganic Arsenic Induces Elevated p53 Levels with Altered Functionality Impacting the Expression of Toll-like Receptor 3 and Other Target Genes in Immortalized Prostate Epithelial Cells"

_ijms, 2025, doi:10.3390/ijms26094253_

Round 1
Reviewer 1 Report
Comments and Suggestions for Authors
The authors discussed "Inorganic arsenic induces elevated p53 levels", and I have some questions to be considered:
(1) What is the significance of using inorganic arsenic?
(2) Any general mechanism for this?
(3) What are the breakthrough points and the next milestone after this investigation?
Author Response
The authors discussed "Inorganic arsenic induces elevated p53 levels", and I have some questions to be considered:
(1) What is the significance of using inorganic arsenic?
We thank the reviewer for these thoughtful questions, which helped us improve the clarity and context of our work. Chronic exposure to inorganic arsenic (iAs) through contaminated drinking water remains a serious global health concern, particularly in regions where groundwater contamination is endemic. iAs has been classified as a Group 1 carcinogen by the International Agency for Research on Cancer (IARC), and epidemiological studies have associated it with increased risk of several malignancies, including prostate cancer. Despite this, the molecular mechanisms by which iAs contributes to prostate carcinogenesis remain poorly understood. Our study addresses this gap using a chronic low-dose exposure model, which more accurately reflects environmental exposure conditions relevant to human populations. This approach allows us to explore early molecular events in prostate epithelial cells under conditions that simulate real-world toxicant exposure.
To clarify this point, we expanded the second paragraph of the Introduction (page 1, lines 43-52 of the Word document and lines 44–51 of the PDF version) to include a more explicit justification for the use of inorganic arsenic and the rationale behind the model employed.
(2) Any general mechanism for this?
We thank the reviewer for this important question. In the revised manuscript, we elaborate on a potential mechanism by which chronic inorganic arsenic (iAs) exposure may disrupt p53 functionality. While iAs is a well-known carcinogen associated with oxidative stress, our study highlights a less explored consequence: p53 accumulation that is not accompanied by functional transcriptional activity.
We propose that chronic exposure to iAs leads to mitochondrial dysfunction and an increase in reactive oxygen species (ROS), which can cause DNA damage but also interfere with the post-translational modifications necessary for p53 activation. These oxidative stress-induced modifications (such as oxidation of cysteine residues or altered phosphorylation/acetylation patterns) may hinder p53’s ability to bind promoter regions and recruit co-activators, even in the absence of TP53 mutations.
To clarify this point, we have added a detailed paragraph in the Discussion (page 11, lines 358–376 of the Word document and lines 337-354 of the PDF version), referencing relevant studies. Although we did not directly assess specific modifications, our ChIP data and the downregulation of canonical p53 target genes (e.g., TLR3, CDKN1A) strongly support the idea of a functionally impaired p53 in iAs-exposed cells. This mechanism broadens our understanding of p53 dysregulation beyond the mutational paradigm and suggests alternative therapeutic avenues in prostate cancer.
(3) What are the breakthrough points and the next milestone after this investigation?
The key conceptual advance of our study is the finding that chronic exposure to inorganic arsenic (iAs) can induce elevated p53 expression while simultaneously impairing its transcriptional activity—even in the absence of TP53 mutations. This challenges the conventional paradigm that associates p53 inactivation primarily with mutational events and highlights a non-genetic mechanism of tumor suppressor dysfunction. Notably, the decoupling of p53 expression from the activation of downstream targets like TLR3 suggests broader implications for immune evasion and therapy resistance in prostate cancer.
The next milestone will be to identify the specific post-translational modifications or chromatin-related changes responsible for this altered p53 activity. We also aim to validate this mechanism in in vivo models and patient-derived tissues, with the long-term goal of informing strategies for therapeutic reactivation of wild-type p53, particularly in the context of TLR3-targeted interventions.
To reflect this more explicitly in the manuscript, we have added a paragraph to the final section of the Discussion (page 11, lines 395–403 of the Word document and lines 372-380 of the PDF version), emphasizing the novelty of our findings and their translational potential.
We thank the reviewer for this insightful comment, which helped us enrich both the discussion and the overall structure of the article.
Reviewer 2 Report
Comments and Suggestions for Authors
The authors reported that the mRNA and protein levels of p53 were upregulated in RWPE-1 and HPrEC after long-term treatment with NaAsO. However, the expression of its potential target genes TLR3 and MDM2 showed a downward trend. The authors attempted to establish a correlation between these two results.
1, The basis for selecting the NaAsO exposure concentration. The method section contains the effect of NaAsO treatment on cell viability, but the results of this section are not shown.
2, Supplementary Figure 1 shows that p53 expression has been on the rise at 1-4 weeks. The reason why the article selected 29 weeks.
3, The article did not analyze the changes in post-translational modifications of p53, such as phosphorylation.
4, The article compares the differential expression of genes such as p53 and TLR3 in various cancers, which is not closely related to the topic of the article and is suspected of making up data. The author may consider removing it to improve the integrity of the article.
Author Response
The authors reported that the mRNA and protein levels of p53 were upregulated in RWPE-1 and HPrEC after long-term treatment with NaAsO. However, the expression of its potential target genes TLR3 and MDM2 showed a downward trend. The authors attempted to establish a correlation between these two results.
1, The basis for selecting the NaAsO exposure concentration. The method section contains the effect of NaAsO treatment on cell viability, but the results of this section are not shown.
We thank the reviewer for pointing out this omission. The concentration of 3 µM NaAsO was selected based on preliminary viability assays showing that, while cytotoxic effects were evident, cell viability remained above 50%, allowing for chronic exposure without total cell loss. This condition is essential for modeling sustained environmental exposure and cellular adaptation, as supported by previous studies such as Tokar et al. (2010).
To address this concern, we have now included the missing viability data as Supplementary Figure 3 and updated the Methods section (page 11, lines 419-426 of the Word document and lines 396-407 of the PDF version) to clarify the rationale behind our selection. These changes explicitly explain how the concentration enables long-term treatment while avoiding complete cytotoxicity, consistent with prior transformation models.
2, Supplementary Figure 1 shows that p53 expression has been on the rise at 1-4 weeks. The reason why the article selected 29 weeks.
We appreciate the reviewer’s interest in the rationale for selecting the 29-week exposure point. While p53 expression begins to rise as early as weeks 1–4, our objective was to model chronic exposure, which is more representative of long-term environmental arsenic exposure in human populations. The selection of 29 weeks was based on previous transformation protocols (Tokar et al., 2011) and allowed us to model the long-term effects of chronic iAs exposure, simulating environmental exposure scenarios. Previous studies suggest that chronic iAs exposure spanning several months better recapitulates the molecular alterations observed in arsenic-associated malignancies. We selected week 29 as a representative late time point after prolonged exposure, where p53 expression is elevated but canonical target activation is impaired.
This rationale is now clarified in the revised Methods section (page 11, lines 424-431 of the Word document and lines 400–407 of the PDF version).
3, The article did not analyze the changes in post-translational modifications of p53, such as phosphorylation.
We agree that this is an important consideration and have clarified it accordingly. While the scope of this study did not include experimental assessment of specific post-translational modifications (PTMs), we do acknowledge this limitation in the revised Discussion section (page 10, lines 363–376 of the Word document and lines 341–354 of the PDF version). Due to resource constraints and our focus on transcriptomic and ChIP-based analyses, we prioritized examining p53 transcriptional output and promoter binding activity. However, our findings—namely, the accumulation of p53 protein alongside decreased expression of canonical p53 targets—strongly suggest a functional impairment that may be mediated by PTMs. We now emphasize this as a central hypothesis and a key direction for future work. By identifying specific PTMs and their upstream regulators, it may become possible to pharmacologically restore p53 function in arsenic-associated prostate cancer, opening new therapeutic opportunities where conventional strategies have been limited.
4, The article compares the differential expression of genes such as p53 and TLR3 in various cancers, which is not closely related to the topic of the article and is suspected of making up data. The author may consider removing it to improve the integrity of the article.
We respectfully and strongly disagree with the concern regarding data integrity. All the results presented in Figure 4 were generated using publicly available RNA-seq datasets from The Cancer Genome Atlas (TCGA), accessed through the OncoDB platform (Tang et al., 2022). The data were retrieved and analyzed using the built-in statistical tools provided by OncoDB, and the differential expression analyses are fully reproducible.
To eliminate any possible ambiguity, we revised the Methods section 4.8 (page 14 lines 542-567 of the Word document and lines 518-534 of the PDF version) to describe in detail how the data were obtained, which statistical methods were used, and how visualizations were generated. This ensures that any reader can replicate our analysis directly from the source.
Furthermore, the selection of adrenocortical carcinoma (ACC) and chromophobe kidney carcinoma (KICH) was not arbitrary. As clarified in the revised Results section 2.4 (page 5 now lines 158–197 of the Word document and lines 156-182 of the PDF version), we selected these tumor types based on their anatomical and developmental proximity to the prostate. Specifically, the adrenal gland, kidney, and prostate all derive from intermediate mesodermal lineages and share common regulatory pathways during urogenital development. This shared origin makes them suitable comparative models to explore whether the p53 dysregulation observed in arsenic-exposed prostate epithelial cells is specific to the prostate or also evident in other tissues with similar developmental backgrounds.
Our intention was not to generalize beyond prostate cancer, but to provide contrastive context supporting the tissue-specific nature of the observed p53 functional uncoupling. The section was rewritten accordingly to reflect this intent and to reinforce the scientific transparency of the analysis.
Reviewer 3 Report
Comments and Suggestions for Authors
The research explores the role of TLR3 activation and its interaction with p53 in the context of iAs exposure. This innovative approach may provide new insights into potential therapeutic targets for advanced PCa, particularly through immunotherapy. The use of prostate epithelial cell lines (RWPE-1 and HPrEC) and the implementation of ChIP assays to assess p53 binding capacity lend credibility to the findings. The research highlights the role of post-translational modifications of p53, such as phosphorylation, in modulating its activity. This focus on the regulatory mechanisms of p53 may pave the way for future studies aimed at restoring its function in cancer therapy.
The discussion could be enhanced if the authors include the limitations of their study based on prostate epithelial cell lines, as well as acknowledge that other factors may influence the expression and activity of p53 under in vivo conditions.
Author Response
Comments and Suggestions for Authors
The research explores the role of TLR3 activation and its interaction with p53 in the context of iAs exposure. This innovative approach may provide new insights into potential therapeutic targets for advanced PCa, particularly through immunotherapy. The use of prostate epithelial cell lines (RWPE-1 and HPrEC) and the implementation of ChIP assays to assess p53 binding capacity lend credibility to the findings. The research highlights the role of post-translational modifications of p53, such as phosphorylation, in modulating its activity. This focus on the regulatory mechanisms of p53 may pave the way for future studies aimed at restoring its function in cancer therapy.
The discussion could be enhanced if the authors include the limitations of their study based on prostate epithelial cell lines, as well as acknowledge that other factors may influence the expression and activity of p53 under in vivo conditions.
Response
We thank the reviewer for this thoughtful suggestion and for the supportive feedback provided throughout the review. In response to this comment, we have added a paragraph to the Discussion section (page 10, lines 377–403 of the Word document and lines 377–403 of the PDF version) explicitly acknowledging the limitations of using immortalized prostate epithelial cell lines. We now also address the potential influence of the tumor microenvironment, stromal interactions, immune components, and systemic factors—elements absent from our in vitro system—that could modulate p53 activity and TLR3 expression in vivo. Furthermore, we emphasize that future studies using patient-derived organoids or in vivo models will be essential to validate and extend our findings within more physiologically relevant contexts.
Round 2
Reviewer 2 Report
Comments and Suggestions for Authors
The author made appropriate revisions to the article according to the suggestions, which improved the overall quality of the article. No further suggestions.